# CELL-DIFF: UNIFIED DIFFUSION MODELING FOR PROTEIN SEQUENCES AND MICROSCOPY IMAGES

## ABSTRACT

Fluorescence microscopy is ubiquitously used in cell biology research to characterize the cellular role of a protein. To help elucidate the relationship between the amino acid sequence of a protein and its cellular function, we introduce CELL-Diff, a unified diffusion model facilitating bidirectional transformations between protein sequences and their corresponding microscopy images. Utilizing reference cell morphology images and a protein sequence, CELL-Diff efficiently generates corresponding protein images. Conversely, given a protein image, the model outputs protein sequences. CELL-Diff integrates continuous and diffusion models within a unified framework and is implemented using a transformer-based network. We train CELL-Diff on the Human Protein Atlas (HPA) dataset and fine-tune it on the OpenCell dataset. Experimental results demonstrate that CELL-Diff outperforms existing methods in generating high-fidelity protein images, making it a practical tool for investigating subcellular protein localization and interactions.

## 1 INTRODUCTION

Protein sequences inherently encode their functions, and predicting these functions solely from sequence information has become a critical area of research. With the development of artificial intelligence, learning-based methods are increasingly employed to predict a wide range of protein properties, including structural conformation (Jumper et al., 2021; Baek et al., 2021), interaction partners (Evans et al., 2021), subcellular localization (Almagro Armenteros et al., 2017; Khwaja et al., 2024b), and binding affinity (Rube et al., 2022). Concurrently, the rapid development of generative models has enabled researchers to design functional proteins (Madani et al., 2023; Dauparas et al., 2022) and drug-like molecules (Isigkeit et al., 2024). These computational methods allow for large-scale virtual screening, significantly reducing the costs and resources associated with experimental validation. The advent of those technologies presents significant opportunities for biomedical research, potentially accelerating advancements in therapeutic target identification, drug discovery, and the investigation of biochemical pathways (Palma et al., 2012).

In this work, we focus on the relationship between protein sequences and their cellular functions as characterized by microscopy images. Specifically, we focus on fluorescence microscopy which is ubiquitously used in nearly all cell biology research. Fluorescence microscopy images provide extremely rich information for proteins of interest in the cellular context, such as their expression level, subcellular distribution, and molecular interactions as can be measured by spatial colocalization. Such information characterizes protein functional activities as well as the physiological and pathological state of cells. Disease-causing genetic mutations can alter the amino acid sequence of proteins, resulting in changes in image phenotypes by modifying gene expression patterns, reshaping molecular interaction profiles, or globally perturbing cellular states. As a first step towards building a model that connects the sequence of proteins and their cellular images, recently, Khwaja et al. (2024b) proposed CELL-E, a text-to-image transformer that predicts fluorescence protein images from sequence input and cell morphology condition. Furthermore, CELL-E2 (Khwaja et al., 2024a) was developed to enhance the image generation speed of CELL-E by utilizing the idea from MaskGIT (Chang et al., 2022). Additionally, CELL-E2 facilitates the bidirectional transformation between sequences and images. However, their image model only allowed output of highly blurred images lacking fine details to discern any of the subcellular structures other than the most prominent one (i.e. the nucleus), restricting their applicability only to the study of a very limited set of sequences features (i.e. the nuclear localization signal).

To expand the application of sequence-to-cell-image generative models, we introduce CELL-Diff, a unified diffusion model that enables bidirectional transformation between protein sequences and their corresponding microscopy images. Specifically, by utilizing cell morphology images including the nucleus and cytoplasmic markers

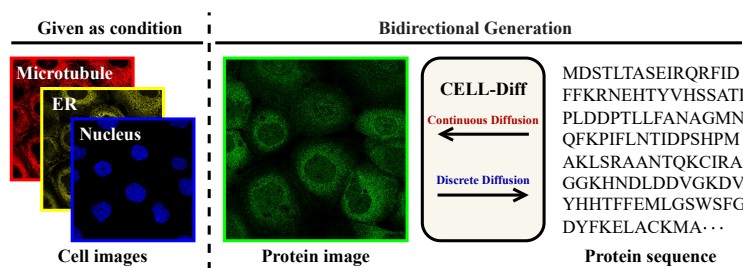

Figure 1: Given cell images as conditional input, CELL-Diff facilitates bidirectional generation between protein sequences and images.

such as endoplasmic reticulum (ER) and microtubule as conditional input, CELL-Diff can generate detailed protein images from given protein sequences. Conversely, it can also output protein sequences when provided with microscopy images, as shown in Figure 1. To enable this bidirectional transformation, CELL-Diff employs the continuous diffusion model for generating microscopy images and the discrete diffusion model for redesigning protein sequences, which can be further integrated within a unified framework. Inspired by Unidiffuser(Bao et al., 2023), we adopt separate diffusion time steps for the continuous and discrete diffusion models, enabling conditional generation. The final objective function comprises the noise prediction loss for the continuous diffusion model and the masked value prediction loss for the discrete diffusion model. Moreover, we design an attention-based U-Net model (Ronneberger et al., 2015; Peebles & Xie, 2023) to integrate information from both modalities efficiently. We evaluate CELL-Diff on HPA dataset (Digre & Lindskog, 2021), which provides cellular microscopy images of human proteins based on fixed immunofluorescence staining. Subsequently, we fine-tune the model on the OpenCell dataset (Cho et al., 2022), which offers live microscopy images of different human cell lines, each tagged with a single protein via CRISPR/Cas9 gene editing.

- We present CELL-Diff, a diffusion-based generative model that enables conditional bidirectional generation of protein sequences and their corresponding microscopy images. By integrating the continuous diffusion and discrete diffusion models, CELL-Diff can be trained within a unified framework. We propose an attention-based U-Net model for implementing CELL-Diff, which effectively integrates information from images and sequences.

- We train CELL-Diff on the HPA dataset using different conditional cell images and fine-tune it on the OpenCell dataset. Experimental results show that our model generates more detailed and sharper protein images compared to previous methods.

## 2 RELATED WORKS

Multi-modal generative modeling can be formalized as learning the conditional or joint distribution between modalities. Representative applications include text-to-image generation (Ramesh et al., 2021; Ding et al., 2021; Nichol et al., 2022), image-to-text generation (image captioning) (Mokady et al., 2021; Chen et al., 2023), text-to-video generation (Ho et al., 2022), and text-to-speech (Chen et al., 2021; Popov et al., 2021). Most of these approaches rely on diffusion models or auto-regressive models for the generation and typically focus on unidirectional transformation. However, our goal is to achieve bidirectional generation, which requires the learning of joint distributions. To achieve this, Hu et al. (2023) proposed a discrete diffusion-based model for learning the joint distribution between images and text, though its scalability remains unexplored. Bao et al. (2023) introduced Unidiffuser, a unified diffusion model capable of unconditional, conditional, and joint generation. The key observation of Unidiffuser is that the learning objective of the diffusion score function can be unified in a general framework with multiple diffusion time steps. Furthermore, Zhou et al. (2024) developed Transfusion, which integrates auto-regressive and diffusion models for both single and cross-modality generation. Transfusion combines the auto-regressive loss with diffusion, training a single transformer model using an extended causal mask. These methods generally depend on large pre-trained encoders for images and text, such as CLIP (Radford et al., 2021) and VQGAN (Esser et al., 2021). However, for microscopy images, the variability in equipment and experimental conditions limits the availability of such robust image encoders, making the direct

application of these models challenging. Indeed, the two previous protein-sequence-to-microscopy generators, CELL-E (Khwaja et al., 2024b) and CELL-E2 (Khwaja et al., 2024a), which both used VQGAN, only produce coarse-grain images that have too much blur to distinguish fine-scale sub-cellular structures such as cytoskeleton. As for CELL-Diff, we combine continuous and discrete diffusion to enable bidirectional transformation between protein images and sequences. The model is trained on the pixel space, offering a straightforward and efficient approach.

## 3 TECHNICAL BACKGROUND

Before delving into our unified diffusion model, we briefly introduce the background of diffusion models applicable to continuous and discrete state spaces. Specifically, we employ the continuous diffusion model for microscopy images and the discrete diffusion model for protein sequences.

### 3.1 DIFFUSION MODEL FOR CONTINUOUS STATE SPACES

Let $\mathbf{I}_0$ be a continuous random variable in $\mathbb{R}^d$, where $d$ denotes the dimension, and let $\mathbf{I}_{1:T} = \{\mathbf{I}_t\}_{t=1}^T$ be a sequence of latent variables, with $t$ as the index for diffusion steps. The diffusion model involves two processes: the forward process and the reverse process. In the forward process, the diffusion model progressively injects noise into the initial data $\mathbf{I}_0$, transforming it into a Gaussian random variable $\mathbf{I}_T$. In the reverse process, the model learns to invert the diffusion process through a denoising model and generate new data by gradually eliminating the noise.

**Forward process.** The forward process involves injecting noise into the initial data. Given a variance schedule $\{\beta_t\}_{t=1}^T$, the forward process is defined as:

$$q(\mathbf{I}_t|\mathbf{I}_{t-1}) = \mathcal{N}(\sqrt{1-\beta_t}\mathbf{I}_{t-1}, \beta_t \boldsymbol{I}_d), \quad t = 1, \ldots, T. \tag{1}$$

Let $\alpha_t = 1 - \beta_t$ and $\bar{\alpha}_t = \prod_{i=1}^t \alpha_i$. Then, for any arbitrary step $t$, it holds that $q(\mathbf{I}_t|\mathbf{I}_0) = \mathcal{N}(\sqrt{\bar{\alpha}_t}\mathbf{I}_0, (1-\bar{\alpha}_t)\boldsymbol{I}_d)$. Consequently, for a sufficiently large $T$, this process will transform $\mathbf{I}_0$ into an isotropic Gaussian variable.

**Reverse process.** The goal of the reverse process is to generate new samples from $p(\mathbf{I}_0)$ starting from a Gaussian random variable $\mathbf{I}_T \sim \mathcal{N}(0, \boldsymbol{I}_d)$. The reverse process is defined by a Markov Chain with trainable transitions:

$$p_\theta(\mathbf{I}_{t-1}|\mathbf{I}_t) = \mathcal{N}(\mu_\theta(\mathbf{I}_t, t), \sigma_t^2 \boldsymbol{I}_d), \quad t = 1, \ldots, T. \tag{2}$$

Here, $\mu_\theta$ represents parameterized neural networks designed to estimate the means from the current state, and $\sigma_t^2$ denotes the variance.

**Training objective.** The training objective function can be derived using variational inference. Instead of optimizing the intractable log-likelihood function $\log p(\mathbf{I}_0)$, the diffusion model maximize its ELBO:

$$\mathbb{E}_{q(\mathbf{I}_{1:T}|\mathbf{I}_0)}\left[\log p_\theta(\mathbf{I}_0|\mathbf{I}_1) - D_{\mathrm{KL}}(q(\mathbf{I}_T|\mathbf{I}_0)\|p_\theta(\mathbf{I}_T)) - \sum_{t=2}^T D_{\mathrm{KL}}\left(q(\mathbf{I}_{t-1}|\mathbf{I}_t, \mathbf{I}_0)\|p_\theta(\mathbf{I}_{t-1}|\mathbf{I}_t)\right)\right], \tag{3}$$

where $q(\mathbf{I}_{t-1}|\mathbf{I}_t, \mathbf{I}_0)$ has an formulation as $\mathcal{N}(\frac{\sqrt{\bar{\alpha}_{t-1}}\beta_t}{1-\bar{\alpha}_t}\mathbf{I}^0 + \frac{\sqrt{\alpha_t}(1-\bar{\alpha}_{t-1})}{1-\bar{\alpha}_t}\mathbf{I}^t, \frac{(1-\bar{\alpha}_{t-1})\beta_t}{1-\bar{\alpha}_t}\boldsymbol{I}_d)$.

To simplify the computation, Ho et al. (2020) used a training objective based on a variant of the ELBO in Equation 3 as

$$L_{\mathrm{DDPM}} = \mathbb{E}_{t,\mathbf{I}_0,\epsilon}\|\epsilon_\theta(\sqrt{\bar{\alpha}_t}\mathbf{I}_0 + \sqrt{1-\bar{\alpha}_t}\epsilon, t) - \epsilon\|_2^2, \tag{4}$$

where $\epsilon \sim \mathcal{N}(\mathbf{0}, \boldsymbol{I}_d)$ and $\epsilon_\theta$ is a noise prediction network.

### 3.2 DIFFUSION MODEL FOR DISCRETE STATE SPACES

Several distinct diffusion models are designed for discrete data (Austin et al., 2021; Hoogeboom et al., 2021). This section focuses on the order-agnostic Autoregressive Diffusion Models (OA-ARDM) (Hoogeboom et al., 2022).

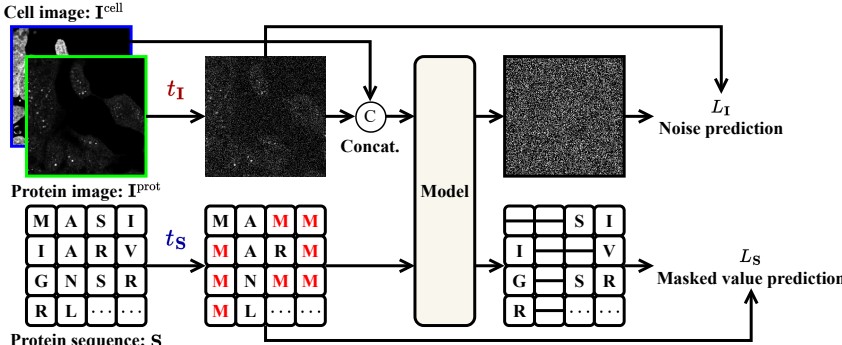

Figure 2: Training losses of CELL-Diff. During each training iteration, the protein image $\mathbf{I}^{\mathrm{port}}$ and sequence $\mathbf{S}$ are transformed using the forward processes of the continuous and discrete diffusion models, with randomly sampled time steps $t_{\mathbf{I}}$ and $t_{\mathbf{S}}$, respectively. The network model is tasked with predicting the noise in the protein image and the masked values from the protein sequence, corresponding to the noise prediction loss $L_{\mathbf{I}}$ and the masked value prediction loss $L_{\mathbf{S}}$.

Let $\mathbf{S} = (\mathbf{S}_1, \ldots, \mathbf{S}_D)$ be a multivariate random variable, where $\forall t \in \{1, \ldots, D\}, \mathbf{S}_t \in \{1, \ldots, K\}$ with $K$ categories. Denote $S_D$ as the set of all permutations of the integers $1, \ldots, D$, and assume $\sigma$ represents a random ordering in $S_D$. Applying Jensen's inequality, we obtain:

$$\log p(\mathbf{S}) = \log \mathbb{E}_{\sigma \sim \mathcal{U}(S_D)} p(\mathbf{S}|\sigma) \geq \mathbb{E}_{\sigma \sim \mathcal{U}(S_D)} \log p(\mathbf{S}|\sigma), \qquad (5)$$

where $\mathcal{U}(S_D)$ denotes the uniform distribution over $S_D$. Following order $\sigma$, $\log p(\mathbf{S}|\sigma)$ can be factorized as $\sum_{t=1}^{D} \log p(\mathbf{S}_{\sigma(t)}|\mathbf{S}_{\sigma(<t)})$, where $\mathbf{S}_{\sigma(<t)} = (\mathbf{S}_{\sigma(1)}, \ldots, \mathbf{S}_{\sigma(t-1)})$. Combining this with Equation 5, we have:

$$\log p(\mathbf{S}) \geq \mathbb{E}_{\sigma \sim \mathcal{U}(S_D)} \sum_{t=1}^{D} \log p(\mathbf{S}_{\sigma(t)}|\mathbf{S}_{\sigma(<t)}) = \mathbb{E}_{\sigma \sim \mathcal{U}(S_D)} \sum_{t=1}^{D} \frac{1}{D-t+1} \sum_{k \in \sigma(\geq t)} \log p(\mathbf{S}_k|\mathbf{S}_{\sigma(<t)}).$$
$$(6)$$

Therefore, denote $\mathbf{f}_\theta$ as the neural network, $\mathcal{C}$ as the categorical distribution, the loss function for OA-ARDM is

$$L_{\text{OA-ARDM}} = \mathbb{E}_{\sigma \sim \mathcal{U}(S_D), t \sim \mathcal{U}(1, \ldots, D)} \frac{1}{D-t+1} \sum_{k \in \sigma(\geq t)} -\log \mathcal{C}(\mathbf{S}_k|\mathbf{f}_\theta(\mathbf{S}_{\sigma(<t)})). \qquad (7)$$

This objective function corresponds to the "Masked Language Modeling" training objective proposed in BERT (Kenton & Toutanova, 2019) with a reweighting term. At each training step, we first sample a time step $t$ from $\mathcal{U}(1, \ldots, D)$, followed by a random ordering $\sigma$ from $\mathcal{U}(S_D)$. We then input $\mathbf{S}_{\sigma(<t)}$ into the model, which predicts the remaining values $\mathbf{S}_{\sigma(\geq t)}$. In the generation step, we first sample a random ordering and then generate the values according to that order. These processes are facilitated through a masking operation, see Appendix A for the details.

# 4 METHODOLOGY

In this section, we introduce our unified diffusion model for generating microscopy images and protein sequences. Let $\mathbf{I}^{\mathrm{prot}}$ represents the protein image, $\mathbf{I}^{\mathrm{cell}}$ represents the cell morphology image, and $\mathbf{S}$ represents the protein sequence. The task of protein image prediction involves sampling from the conditional distribution $p(\mathbf{I}^{\mathrm{prot}}|\mathbf{S}, \mathbf{I}^{\mathrm{cell}})$, while the task of sequence generation involves sampling from $p(\mathbf{S}|\mathbf{I}^{\mathrm{prot}}, \mathbf{I}^{\mathrm{cell}})$. To achieve these goals within a unified diffusion model, we choose to estimate the joint distribution $p(\mathbf{I}^{\mathrm{prot}}, \mathbf{S}|\mathbf{I}^{\mathrm{cell}}))$, which involves model a continues variable $\mathbf{I}^{\mathrm{prot}}$ and a discrete variable $\mathbf{S}$.

## 4.1 PROPOSED METHOD

Let $\mathbf{I}_0$ represent the protein image $\mathbf{I}^{\mathrm{port}}$ and temporarily ignore the cell image $\mathbf{I}^{\mathrm{cell}}$, we consider modeling the joint distribution $p(\mathbf{I}_0, \mathbf{S})$. Following the diffusion models described in in Section 3,

we introduce a sequence of latent variables $\mathbf{I}_{1:T} = \{\mathbf{I}_t\}_{t=1}^{T}$ for $\mathbf{I}_0$ and a random ordering $\sigma \in S_D$ for $\mathbf{S}$. The log-likelihood function satisfies:

$$\log p(\mathbf{I}_0, \mathbf{S}) = \log \mathbb{E}_{\sigma \sim \mathcal{U}(S_D)} \mathbb{E}_{q(\mathbf{I}_{1:T}|\mathbf{I}_0)} \frac{p(\mathbf{I}_{0:T}, \mathbf{S}|\sigma)}{q(\mathbf{I}_{1:T}|\mathbf{I}_0)} \geq \mathbb{E}_{\sigma \sim \mathcal{U}(S_D)} \mathbb{E}_{q(\mathbf{I}_{1:T}|\mathbf{I}_0)} \log \frac{p(\mathbf{I}_{0:T}, \mathbf{S}|\sigma)}{q(\mathbf{I}_{1:T}|\mathbf{I}_0)}. \tag{8}$$

Assuming the same forward and reverse process in Section 3.1, $q(\mathbf{I}_{1:T}|\mathbf{I}_0)$ can be decomposed as $q(\mathbf{I}_T|\mathbf{I}_0) \prod_{t=2}^{T} q(\mathbf{I}_{t-1}|\mathbf{I}_t, \mathbf{I}_0)$.

Regrading $\log p(\mathbf{I}_{0:T}, \mathbf{S}|\sigma)$, the decomposition depends on the factorization order between $\mathbf{I}_{0:T}$ and $\mathbf{S}$. Given a specific factorization order, for sequence $\mathbf{S}$, each decomposition term can be represented as $\log p(\mathbf{S}_{\sigma(t_{\mathbf{S}})}|\mathbf{S}_{\sigma(<t_{\mathbf{S}})}, \mathbf{I}_{\geq t_{\mathbf{I}}})$, where $t_{\mathbf{S}} \in \{1, \ldots, D\}$ and $t_{\mathbf{I}} \in \{0, \ldots, T\}$[1]. Furthermore, since the forward process shown in Equation 1 indicates that the information from $\mathbf{I}_0$ to $\mathbf{I}_T$ is progressively decreasing, we assume $\log p(\mathbf{S}_{\sigma(t_{\mathbf{S}})}|\mathbf{S}_{\sigma(<t_{\mathbf{S}})}, \mathbf{I}_{\geq t_{\mathbf{I}}}) = \log p(\mathbf{S}_{\sigma(t_{\mathbf{S}})}|\mathbf{S}_{\sigma(<t_{\mathbf{S}})}, \mathbf{I}_{t_{\mathbf{I}}})$. For image $\mathbf{I}$, using the Markov Chain model, each decomposition term can be represented as $\log p(\mathbf{I}_T|\mathbf{S}_{\sigma(<t_{\mathbf{S}})})$ and $\log p(\mathbf{I}_{t_{\mathbf{I}}-1}|\mathbf{I}_{t_{\mathbf{I}}}, \mathbf{S}_{\sigma(<t_{\mathbf{S}})})$, where $t_{\mathbf{I}} \in \{1, \ldots, T\}$ and $t_{\mathbf{S}} \in \{1, \ldots, D+1\}$. Combining this with $q(\mathbf{I}_{1:T}|\mathbf{I}_0)$, the KL term in Equation 3 can be expressed as $D_{\mathrm{KL}}\left(q(\mathbf{I}_{t_{\mathbf{I}}-1}|\mathbf{I}_{t_{\mathbf{I}}}, \mathbf{I}_0)\|p(\mathbf{I}_{t_{\mathbf{I}}-1}|\mathbf{I}_{t_{\mathbf{I}}}, \mathbf{S}_{\sigma(<t_{\mathbf{S}})})\right)$, which has a closed-form formulation with Gaussian parameterization.

In practice, the choice of factorization order between $\mathbf{I}_{0:T}$ and $\mathbf{S}$ depends on the downstream purpose. In our case, we aim to generate samples from two conditional distributions $p(\mathbf{I}|\mathbf{S})$ and $p(\mathbf{S}|\mathbf{I})$, which requires simultaneously factorizing $p(\mathbf{I}_{0:T}, \mathbf{S}|\sigma)$ from $\mathbf{I}_{0:T}$ to $\mathbf{S}$ and from $\mathbf{S}$ to $\mathbf{I}_{0:T}$. To achieve this goal, we adopt the approach from UniDiffuser (Bao et al., 2023), considering all possible factorization combinations. Therefore, we maximize the following objective function:

$$\mathbb{E}_{\sigma \sim \mathcal{U}(S_D)} \mathbb{E}_{q(\mathbf{I}_{1:T}|\mathbf{I}_0)} \sum_{t_{\mathbf{S}}=1}^{D} \sum_{t_{\mathbf{I}}=0}^{T} \log p(\mathbf{S}_{\sigma(t_{\mathbf{S}})}|\mathbf{S}_{\sigma(<t_{\mathbf{S}})}, \mathbf{I}_{t_{\mathbf{I}}}) - \sum_{t_{\mathbf{S}}=1}^{D+1} D_{\mathrm{KL}}\left(q(\mathbf{I}_T|\mathbf{I}_0)\|p(\mathbf{I}_T|\mathbf{S}_{\sigma(<t_{\mathbf{S}})})\right)$$

$$+ \sum_{t_{\mathbf{S}}=1}^{D+1} \log p(\mathbf{I}_0|\mathbf{I}_1, \mathbf{S}_{\sigma(<t_{\mathbf{S}})}) - \sum_{t_{\mathbf{S}}=1}^{D+1} \sum_{t_{\mathbf{I}}=1}^{T} D_{\mathrm{KL}}\left(q(\mathbf{I}_{t_{\mathbf{I}}-1}|\mathbf{I}_{t_{\mathbf{I}}}, \mathbf{I}_0)\|p(\mathbf{I}_{t_{\mathbf{I}}-1}|\mathbf{I}_{t_{\mathbf{I}}}, \mathbf{S}_{\sigma(<t_{\mathbf{S}})})\right), \tag{9}$$

where the first term corresponds to the objective function of OA-ARDM in Equation 6, while the remaining terms correspond to the objective function of DDPM in Equation 3.

Utilizing the same parametrization technique as shown in Equation 4 and Equation 7, and considering modeling the joint distribution $p(\mathbf{I}^{\mathrm{port}}, \mathbf{S}|\mathbf{I}^{\mathrm{cell}})$, let $\mathbf{f}_\theta$ denotes the neural network. The training objective function for protein sequence $\mathbf{S}$ is:

$$L_{\mathbf{S}} = \mathbb{E}_{\sigma \sim \mathcal{U}(S_D), \mathbf{I}^{\mathrm{port}}, t_{\mathbf{I}}, t_{\mathbf{S}}, \epsilon} \frac{\sum_{k \in \sigma(\geq t_{\mathbf{S}})} - \log \mathcal{C}(\mathbf{S}_k|\mathbf{f}_\theta(\mathbf{S}_{\sigma(<t_{\mathbf{S}})}, \sqrt{\bar{\alpha}_{t_{\mathbf{I}}}}\mathbf{I}^{\mathrm{port}} + \sqrt{1-\bar{\alpha}_{t_{\mathbf{I}}}}\epsilon, t_{\mathbf{I}}, \mathbf{I}^{\mathrm{cell}}))}{D - t_{\mathbf{S}} + 1}. \tag{10}$$

The training objective function for protein image $\mathbf{I}^{\mathrm{port}}$ is:

$$L_{\mathbf{I}} = \mathbb{E}_{\sigma \sim \mathcal{U}(S_D), \mathbf{I}^{\mathrm{port}}, t_{\mathbf{I}}, t_{\mathbf{S}}, \epsilon} \|\mathbf{f}_\theta(\mathbf{S}_{\sigma(<t_{\mathbf{S}})}, \sqrt{\bar{\alpha}_{t_{\mathbf{I}}}}\mathbf{I}^{\mathrm{port}} + \sqrt{1-\bar{\alpha}_{t_{\mathbf{I}}}}\epsilon, t_{\mathbf{I}}, \mathbf{I}^{\mathrm{cell}}) - \epsilon\|_2^2. \tag{11}$$

In summary, combining Equation 10 and Equation 11 and introduce a balancing coefficient $\lambda$, the total loss of the proposed CELL-Diff model is:

$$L_{\text{CELL-Diff}} = L_{\mathbf{S}} + \lambda L_{\mathbf{I}}. \tag{12}$$

The training strategy is shown in Figure 2.

## 4.2 MODEL DETAILS

**Inference.** After training, we can generate samples from two conditional distributions: $p(\mathbf{I}^{\mathrm{port}}|\mathbf{S}, \mathbf{I}^{\mathrm{cell}})$ and $p(\mathbf{S}|\mathbf{I}^{\mathrm{port}}, \mathbf{I}^{\mathrm{cell}})$. Specifically, to generate the protein image $\mathbf{I}^{\mathrm{port}}$, we utilize the conventional reverse diffusion process as shown in Equation 2, conditioning on the unmasked protein

---

[1]Given that the gap between $\mathbf{I}_T$ and the standard Gaussian noise is negligible, we assume $\log p(\mathbf{S}_{\sigma(t_{\mathbf{S}})}|\mathbf{S}_{\sigma(<t_{\mathbf{S}})}) = \log p(\mathbf{S}_{\sigma(t_{\mathbf{S}})}|\mathbf{S}_{\sigma(<t_{\mathbf{S}})}, \mathbf{I}_T)$.

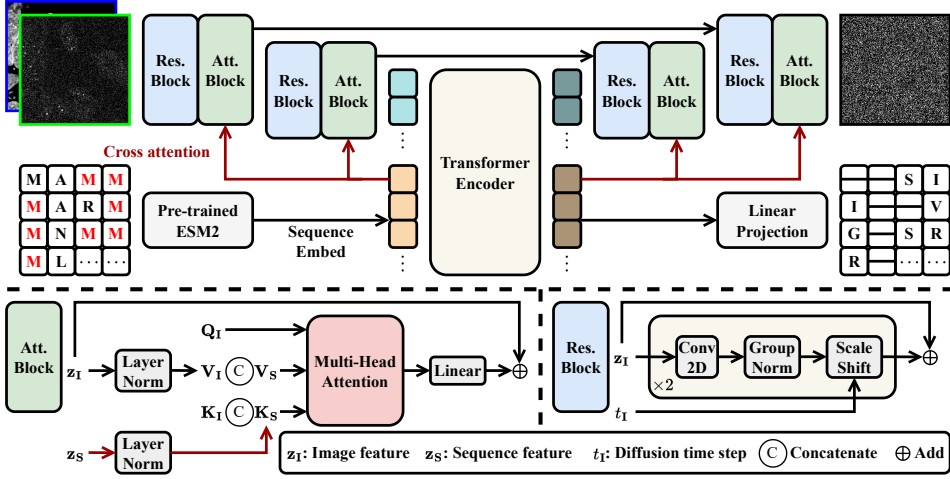

Figure 3: Network architecture of CELL-Diff. Microscopy images are embedded into a latent sequence through residual and attention blocks. The protein sequences are embedded using a pretrained ESM2 model (Lin et al., 2022). These embeddings are concatenated and processed by a transformer model. The U-Net architecture (Ronneberger et al., 2015) is employed to output the noise in the protein image, while a linear projection is utilized to predict the masked values in the protein sequence. Cross-attention mechanisms are implemented to enhance information integration from images and sequences.

sequence $\mathbf{S}$ and the cell image $\mathbf{I}^{\text{cell}}$. The network model employed for generation is $\mathbf{f}_\theta(\mathbf{S}, \cdot, t_\mathbf{I}, \mathbf{I}^{\text{cell}})$, where $t_\mathbf{I} = 1, \ldots, T$. For the generation of the protein sequence $\mathbf{S}$, we utilize the reverse process of discrete diffusion OA-ARDM (Hoogeboom et al., 2022). We first sample a random ordering $\sigma$, and then generate sequence from $p(\mathbf{S}_{\sigma(t_\mathbf{S})}|\mathbf{S}_{\sigma(<t_\mathbf{S})}, \mathbf{I}^{\text{port}}, \mathbf{I}^{\text{cell}})$, where $t_\mathbf{S} = 1, \ldots, D$. The network model in this scenario is $\mathbf{f}_\theta(\cdot, \mathbf{I}^{\text{port}}, 0, \mathbf{I}^{\text{cell}})$. The sampling algorithm for OA-ARDM is shown in Algorithm 2.

**Network architecture.** As shown in Equation 11 and Equation 10, the network model $\mathbf{f}_\theta$ takes four inputs: the protein sequence $\mathbf{S}$, the protein image $\mathbf{I}^{\text{port}}$, the cell image $\mathbf{I}^{\text{cell}}$, and the diffusion time step $t_\mathbf{I}$. To process the protein and cell images, we first concatenate them and then apply the commonly used U-Net architecture (Ronneberger et al., 2015). The concatenated images are fed into a series of downsampling blocks, transforming into image embeddings. The protein sequences are embedded using a pre-trained ESM2 model Lin et al. (2022). Then, the image and protein embeddings are concatenated and processed using an encoder-only transformer model. After passing through the transformer module, the concatenated feature tensors are split into image and sequence feature tensors. The image feature tensors are then upsampled and combined with the downsampling features to output the noise from the protein image. The sequence feature tensor is processed using a linear projector to predict the masked values. The upsampling and downsampling blocks in the U-Net consist of residual and attention blocks. To enhance the integration of sequence information within the image processing component, we utilize cross-attention mechanisms with the attention blocks. Furthermore, we employ the adaptive layer norm zero (adaLN-Zero) conditioning method (Peebles & Xie, 2023) for incorporating the diffusion time step $t_\mathbf{I}$. The network architecture is illustrated in Figure 3.

## 5 EXPERIMENTS

### 5.1 DATASETS

**Human Protein Atlas.** The Human Protein Atlas (HPA) dataset (Digre & Lindskog, 2021) includes immunofluorescence images across various human cell lines with the proteins of interest stained by antibodies. It provides cellular images for 12,833 proteins, as well as corresponding cell morphology images consisting of staining for the nucleus, ER, and microtubules. For each protein, the dataset

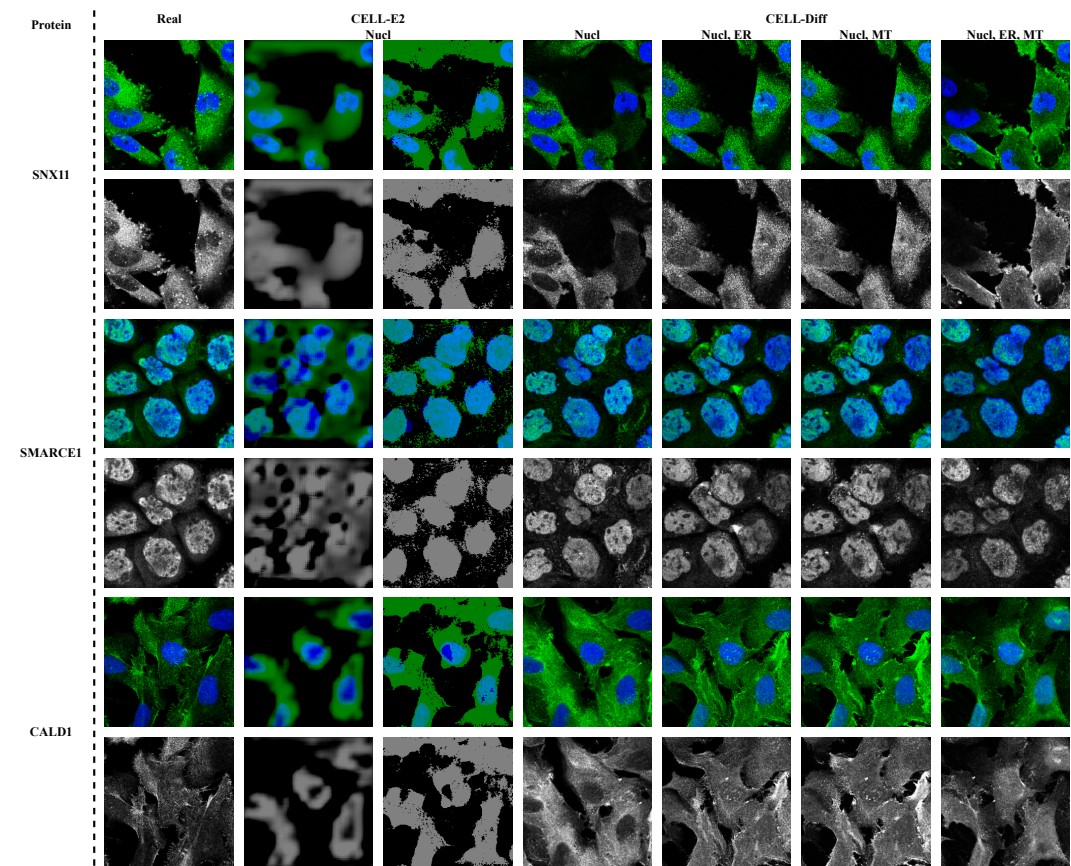

Figure 4: Visual results of protein image generation on HPA dataset.

includes multiple microscopy images from different cell lines. The corresponding protein sequences can be accessed from the UniProt dataset (UniProt Consortium, 2018). In total, we have collected 88,483 data points, each containing a protein sequence, a protein image, a nucleus image, an ER image, and a microtubule image.

**OpenCell.** The OpenCell (Cho et al., 2022) dataset provides a library of 1,311 CRISPR-edited HEK293T human cell lines, each with a target protein fluorescently tagged using the split-mNeonGreen2 system. For each target protein, OpenCell provides 4–5 confocal images along with a reference nucleus image. The cells were imaged live, offering a more accurate representation of protein distribution than the immunofluorescence images from HPA. Notably, 1,102 proteins are common between the HPA and OpenCell datasets. In total, we collected 6,301 data points, each containing a protein sequence, a protein image, and a nucleus image.

Given the size limitations of the HPA and OpenCell datasets, particularly in the diversity of protein sequences, we randomly selected 100 proteins from the shared subset between the two datasets as the test set, leaving the remainder for training. The test set for HPA and OpenCell contains 766 and 470 data points, respectively.

## 5.2 IMPLEMENTATION DETAILS

We first train CELL-Diff models on the HPA dataset and then fine-tune on the OpenCell dataset. Both pre-training and fine-tuning are conducted for 100,000 iterations using the Adam optimizer Kingma & Ba (2014). The learning rate is initialized using a linear warm-up strategy, increasing from 0 to $3 \times 10^{-4}$ over the first 1,000 iterations, followed by a linear decay to zero. The batch size is set to 192. For images from the HPA dataset, we apply the random crop of size 1024,

Table 1: Comparison of protein image generation performance on HPA and OpenCell datasets. "Nucl" denotes the nucleus image, "ER" denotes the endoplasmic reticulum image, and "MT" denotes the microtubule image. "FID-T" indicates the FID computed using the thresholded protein image, and "FID-O" indicates the FID computed using the original protein image.

| Dataset | Method | Cell image | MSF-resolvability (nm) ↓ | IoU ↑ | FID-T ↓ | FID-O ↓ |
|---------|--------|------------|--------------------------|-------|---------|---------|
| HPA | CELL-E2 | Nucl | 1872 | 0.461 | 77.0 | 167.0 |
| | **CELL-Diff** | Nucl | 646 | 0.448 | 35.9 | 31.9 |
| | | Nucl, ER | 642 | 0.619 | **32.4** | 25.2 |
| | | Nucl, MT | 642 | 0.601 | 33.0 | 27.1 |
| | | Nucl, ER, MT | **641** | **0.623** | 34.1 | **24.1** |
| OpenCell | CELL-E2 | Nucl | 1239 | 0.515 | 70.4 | 248.1 |
| | **CELL-Diff** | Nucl | **628** | **0.524** | **40.4** | **20.0** |

followed by resizing to 256. For the OpenCell dataset, images are randomly cropped to a size of 256. Data augmentation is performed using random flips and rotations. The sequence embedding dimension is 640, and the transformer module consists of 24 layers with 8-head attention. The U-Net architecture includes three groups of downsampling and upsampling modules, each containing two residual and attention blocks, with channel sizes increasing from 64 to 512. To convert images into sequences, we use the patchify operation from DiT (Peebles & Xie, 2023) with a patch size of 8. CELL-Diff is trained with 1,000 diffusion steps using the shifted cosine noise schedules (Hoogeboom et al., 2023), and use DDIM (Song et al., 2020) with 100 steps to accurate the sampling speed. The weighting coefficient $\lambda$ in Equation 12 is set to 100, and the maximum protein sequence length is 2,048. All models are trained using two Nvidia H100 GPUs.

## 5.3 PROTEIN IMAGE GENERATION

We evaluate the protein image generation performance of CELL-Diff. Given that the protein image prediction problem is relatively new, we compare CELL-Diff with the most closely related method, CELL-E2 (Khwaja et al., 2024a). To provide a quantitative comparison, we introduce the Maximum Spatial Frequency (MSF) resolvability for microscopy images to measure its capability to discern fine structural details. Given a microscopy image $\mathbf{I}$, we define the Fourier Ring Power Spectral Density (FRPSD) as $\text{FRPSD}(r) = \sum_{r_i \in r} |\hat{\mathbf{I}}_1(r_i)|^2$, where $\hat{\mathbf{I}}$ denotes the Fourier transform of $\mathbf{I}$ and $r_i$ denotes the pixel element at radius $r$. The MSF-resolvability is then defined as:

$$\text{MSF-resolvability} = \frac{1}{f}, \ f = \frac{i}{\text{Image Size} \times \text{Pixel Size}}, \ \text{where} \begin{cases} \text{FRPSD}(r) > 10^{-3}, & r < i \\ \text{FRPSD}(r) < 10^{-3}, & r = i \end{cases}. \tag{13}$$

We also employ the Intersection over Union (IoU) metric, which measures the similarity between two masks and is commonly used in image segmentation tasks. To calculate IoU, we apply median value thresholding to the original protein images to generate binary masks, while for CELL-E2, we use the predicted thresholding images. Additionally, we compute the Fréchet Inception Distance (FID) Heusel et al. (2017) score to evaluate the similarity between the real and predicted images. FID is a learning-based metric that evaluates the quality of images generated by generative models. It measures the similarity between the generated and real images regarding their feature distributions. Lower FID scores indicate that the generated images are more similar to the real images. To compute FID, we concatenate the protein and nucleus images as input. In practice, we compute FID-T and FID-O, representing the FID score based on thresholding and original protein images, respectively. The results are shown in Table 1. The results show that CELL-Diff generated images exhibit better MSF-resolvability than CELL-E2. In particular, the MSF-resolvability for the original HPA and OpenCell data are 640 nm and 426 nm, respectively. The results from CELL-Diff are approaching the resolvability of the original training data, allowing us to discern finer details in protein distribution such as various cytoplasmic organelles. Regarding the prediction accuracy metric IoU, CELL-Diff and CELL-E2 achieve comparable performance when using only the nucleus image as the conditional cell image, which can be greatly improved by incorporating additional cell morphology images, such as those of the ER and microtubules. Regarding the learning-based metric

Table 2: Ablation analysis of cross attention module on HPA dataset. The nucleus image is used as the cell morphology image.

| Method | MSF-resolvability (nm) ↓ | IoU ↑ | FID-T ↓ | FID-O ↓ |
|---|---|---|---|---|
| w/o cross attention | 648 | 0.431 | **33.4** | 40.1 |
| **w cross attention** | **646** | **0.448** | 35.9 | **31.9** |

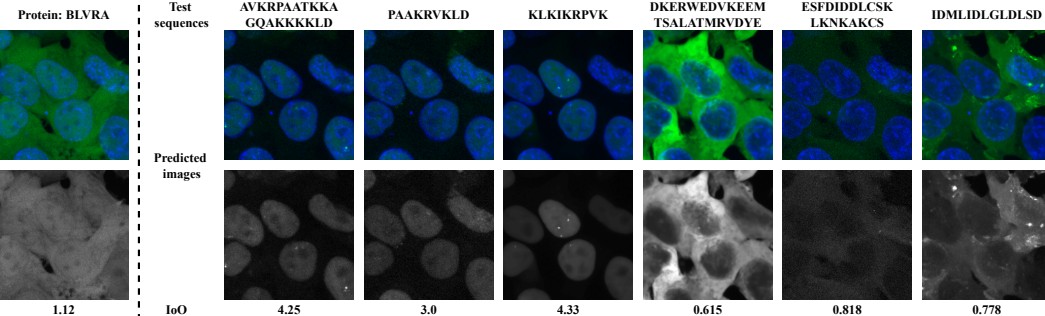

Figure 5: Protein localization signal screening. Test sequences are tagged to the C-terminus of the protein BLVRA. The ratio "IoO" represents the median protein intensity inside the nucleus relative to that outside the nucleus.

FID, CELL-Diff significantly outperforms CELL-E2, further demonstrating the superiority of the proposed method. Visual results are illustrated in Figure 4. From the figure, we find that CELL-Diff accurately predicts protein images from unseen protein sequences. Compared with CELL-E2, CELL-Diff generates more resolvable images, enabling the extraction of more detailed information from the generated images. More results are provided in Appendix B.

## 6 DISCUSSIONS

### 6.1 ABLATION ON CROSS ATTENTION MODULE

We employ the cross-attention mechanism to more effectively integrate information from sequences to images. To evaluate its efficiency, we conduct an ablation analysis of this module on the HPA dataset, see Table 2. The results show that incorporating this module improves most of the quantitative metrics, demonstrating the effectiveness of the cross-attention mechanism.

### 6.2 POTENTIAL APPLICATIONS

In this section, we present three potential applications of the proposed CELL-Diff method for biological discovery. Given that validation relies on biological knowledge and the dataset size is limited, we retrain all models using all the protein sequences from both the HPA and OpenCell datasets.

**Virtual screening of protein localization signal.** CELL-Diff can be applied for the virtual screening of protein localization signals, such as Nuclear Localization Signals (NLS) and Nuclear Export Signals (NES). The NLS is a short amino acid sequence that directs the import of proteins into the nucleus, while the NES facilitates their export from the nucleus. In this approach, the test peptide sequence is tagged to the C-terminus of the protein BLVRA, which is uniformly distributed both inside and outside the nucleus, see the first column of Figure 5. CELL-Diff is then employed to predict the images of the modified protein. The resulting predicted images are analyzed to identify potential localization signals. As illustrated in Figure 5, we compute the median fluorescence intensity inside the nucleus relative to that outside the nucleus, referred to as the IoO ratio. For the original BLVRA protein, the IoO ratio is 1.12. If the IoO ratio of the modified protein exceeds 1.12, the test sequence is likely to function as an NLS, conversely, if the ratio is lower, the sequence is more likely to act as an NES. In Figure 5, we tested known three NLSs and three NESs from the literature. CELL-Diff

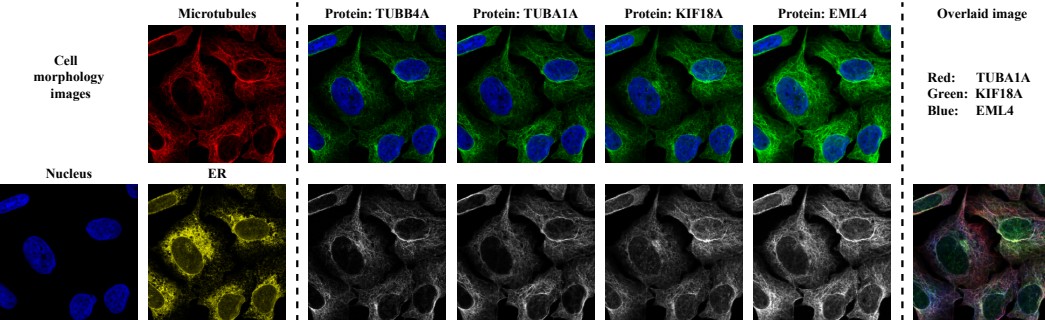

Figure 6: Virtual staining using HPA data. From identical cell morphology images, CELL-Diff generates staining results for various proteins.

successfully recognized these signals, proving its capability as a computational tool for screening potential protein localization signals.

**Virtual staining.** Typical fluorescence microscopes can only fit no more than four color channels in the visible spectrum. Because of this physical limitation, both HPA and OpenCell acquire the images of only one protein of interest per sample, with the other color channels occupied by morphological reference images. Consequently, it is challenging to identify the intracellular spatial relationships among multiple proteins of interest because their images are from different cells. With CELL-Diff, we solve this problem by generating images of these proteins conditioned on the same morphology reference images. These virtual staining images allow the subcellular distributions of an arbitrary number of proteins to be directly compared and potential molecular interactions identified from colocalization, while entirely circumventing the color channel limitation of fluorescence microscopy experiments. We demonstrate that from cell morphology images not in the training data set, CELL-Diff can accurately simulate the imaging results for target protein sequences, see Appendix C. We further demonstrate the use of CELL-Diff to identify molecular interaction by virtually staining two microtubule components (TUBA1A and TUBB4A) and two other proteins, KIF18A and EML4, from the same morphology image. The overlaid image clearly shows the association of KIF18A and EML4 with microtubules in the cell, consistent with their known biological function of microtubule binding, see Figure 6.

**Localization signal generation.** Utilizing image-to-sequence generation, CELL-Diff can be applied to generate novel protein localization signals, such as NLS and NES. Given a cell morphology image and a corresponding protein image, CELL-Diff generates the protein sequences that should be located at the position indicated by the protein image. We started from the Green Fluorescent Protein (GFP) which has no sequence homology with any human proteins and does not contain localization signals by itself (Köhler et al., 1997; Seibel et al., 2007; Kitamura et al., 2015). Conditioned on an image of either a nucleus-localized protein or a nucleus-excluded protein, we used CELL-Diff to append a short peptide either on the N- or C-terminus of GFP. In this way, we generated 200 potential NLS and NES sequences, see Appendix D.

# 7 CONCLUSION

This paper proposes CELL-Diff, a unified diffusion model that facilitates the transformation between protein sequences and microscopy images. Given cell morphology images as conditional inputs, CELL-Diff generates protein images from protein sequences. Conversely, it can generate protein sequences based on microscopy images. The objective function of CELL-Diff is constructed by integrating continuous and discrete diffusion models. Experimental results on the HPA and OpenCell datasets demonstrate that CELL-Diff produces accurate protein images with higher resolvability than previous methods. Potential applications, including virtual screening of protein localization signals, virtual staining, and protein localization signal generation, make CELL-Diff a valuable tool for investigating subcellular protein localization and interactions.

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

# A   IMPLEMENTATION OF DISCRETE DIFFUSION MODEL

The training and sampling process of the discrete diffusion model OA-ARDM (Hoogeboom et al., 2022) can be facilitated through a masking operation. Denote $\mathcal{C}$ as the categorical distribution, the training and sampling algorithms are shown in Algorithm 1 and Algorithm 2, respectively. For each training iteration, we first sample a time step $t$ from $\mathcal{U}(1, \ldots, D)$, and a random ordering $\sigma$ from $\mathcal{U}(S_D)$. Subsequently, we generate a mask $\mathbf{m}$ based on the index $i$ such that $\sigma(i) < t$. We then apply the network $\mathbf{f}_\theta$, which takes $\mathbf{m} \odot \mathbf{S}$ as input, and predicts the masked values $(1 - \mathbf{m}) \odot \mathbf{S}$.

---

**Algorithm 1** Training OA-ARDM

---

**Require:** Network $\mathbf{f}_\theta$, datapoint $\mathbf{S}$.
**Ensure:** $L_{\text{OA-ARDM}}$.
  1: Sample $t \sim \mathcal{U}(1, \ldots, D)$, $\sigma \sim \mathcal{U}(S_D)$.
  2: Compute $\mathbf{m} \leftarrow (\sigma < t)$.
  3: Compute $\mathbf{l} \leftarrow -(1 - \mathbf{m}) \odot \log \mathcal{C}(\mathbf{S}|\mathbf{f}_\theta(\mathbf{m} \odot \mathbf{S}))$.
  4: $L_{\text{OA-ARDM}} \leftarrow \frac{1}{D-t+1}\text{sum}(\mathbf{l})$.

---

**Algorithm 2** Sampling from OA-ARDM

---

**Require:** Network $\mathbf{f}_\theta$.
**Ensure:** Sample $\mathbf{S}$.
  1: Initialize $\mathbf{S} = \mathbf{0}$, sample $\sigma \sim \mathcal{U}(S_D)$.
  2: **for** $t = 0, 1, 2, \ldots, D$ **do**
  3:     $\mathbf{m} \leftarrow (\sigma < t)$ and $\mathbf{n} \leftarrow (\sigma = t)$.
  4:     $\mathbf{S}' \sim \mathcal{C}(\mathbf{S}|\mathbf{f}_\theta(\mathbf{m} \odot \mathbf{S}))$.
  5:     $\mathbf{S} \leftarrow (1 - \mathbf{n}) \odot \mathbf{S} + \mathbf{n} \odot \mathbf{S}$.
  6: **end for**

---

# B   PROTEIN IMAGE GENERATION

We present more protein image generation results. The results on the HPA and OpenCell datasets are shown in Figure 7 and Figure 8, respectively. From these results, we observe that CELL-Diff is capable of generating realistic protein images with high accuracy, enabling the discernment of fine details. Compared to CELL-E2, CELL-Diff produces images with higher resolvability, which provides better clarity of detailed localization structures.

# C   VIRTUAL STAINING RESULTS

Here, we present additional virtual staining results. Figure 9 shows virtual staining using data from the HPA dataset, and Figure 10 shows results using data from the OpenCell dataset. These results demonstrate that CELL-Diff generates accurate staining images compared to real ones, offering an efficient approach for simultaneous visualization of multiple biological features within the same sample.

# D   LOCALIZATION SIGNAL GENERATION

We use CELL-Diff to generate protein localization signals. Specifically, we select the images in Figure 11 as the conditional input for generating NLS and NES signals. Using the CELL-Diff model, we generate short amino acid sequences positioned at the N-terminus (before the GFP sequence) and the C-terminus (after the GFP sequence). A total of 100 potential sequences are generated for each signal type, consisting of 50 N-terminus and 50 C-terminus sequences. Generated NLS and NES sequences are summarized in Table 3 and Table 4, respectively.

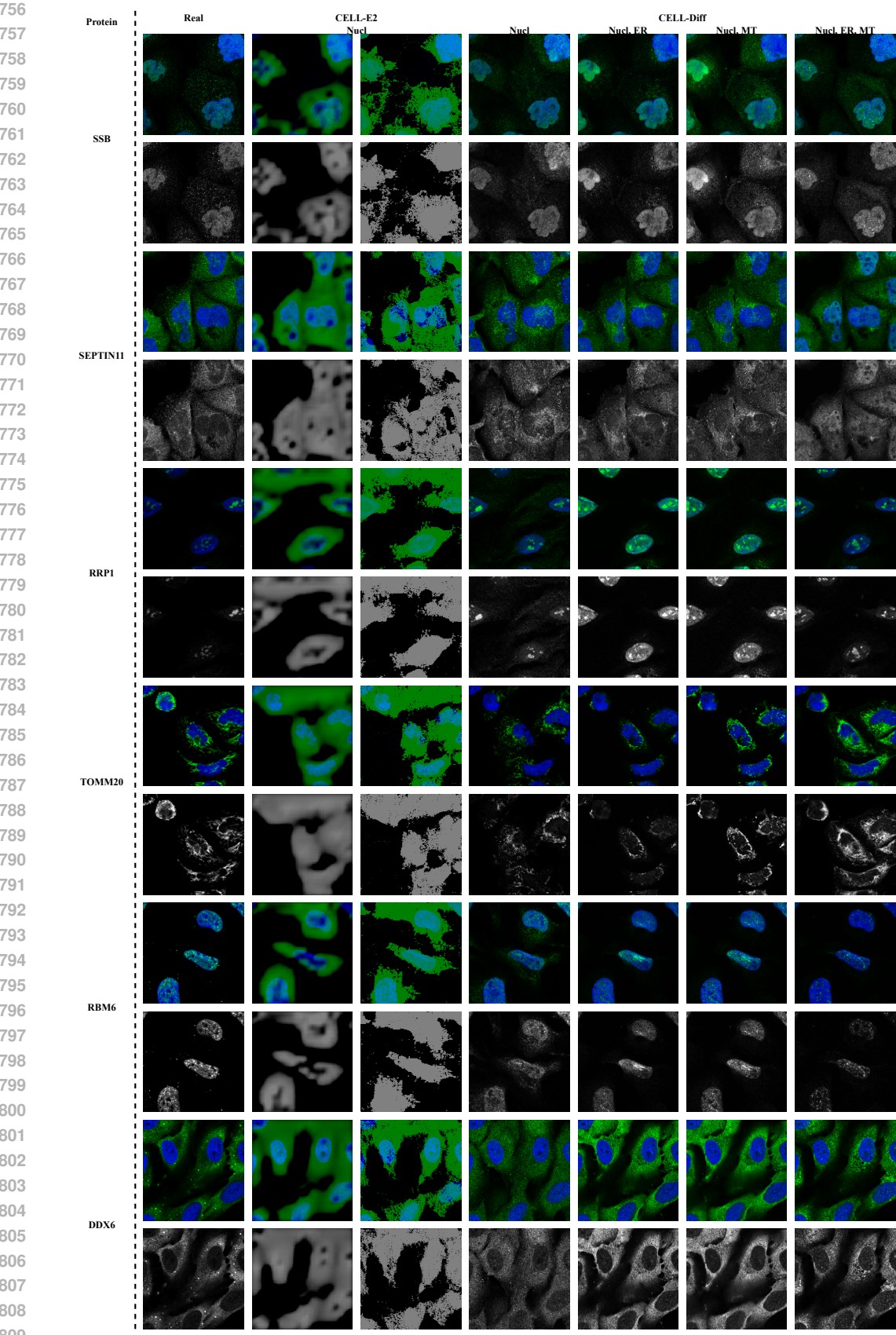

Figure 7: Visual results of protein image generation on HPA dataset.

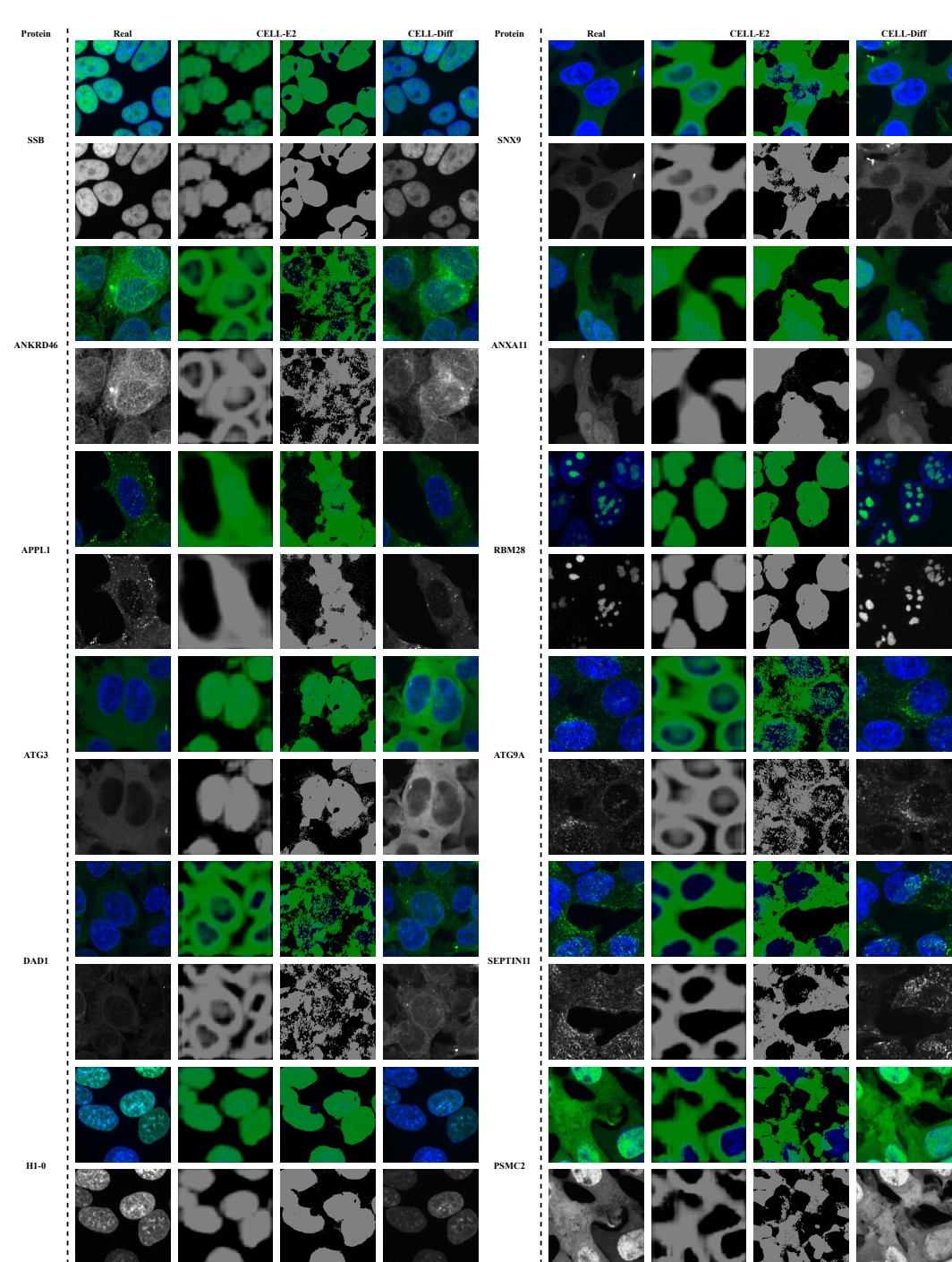

Figure 8: Visual results of protein image generation on OpenCell dataset.

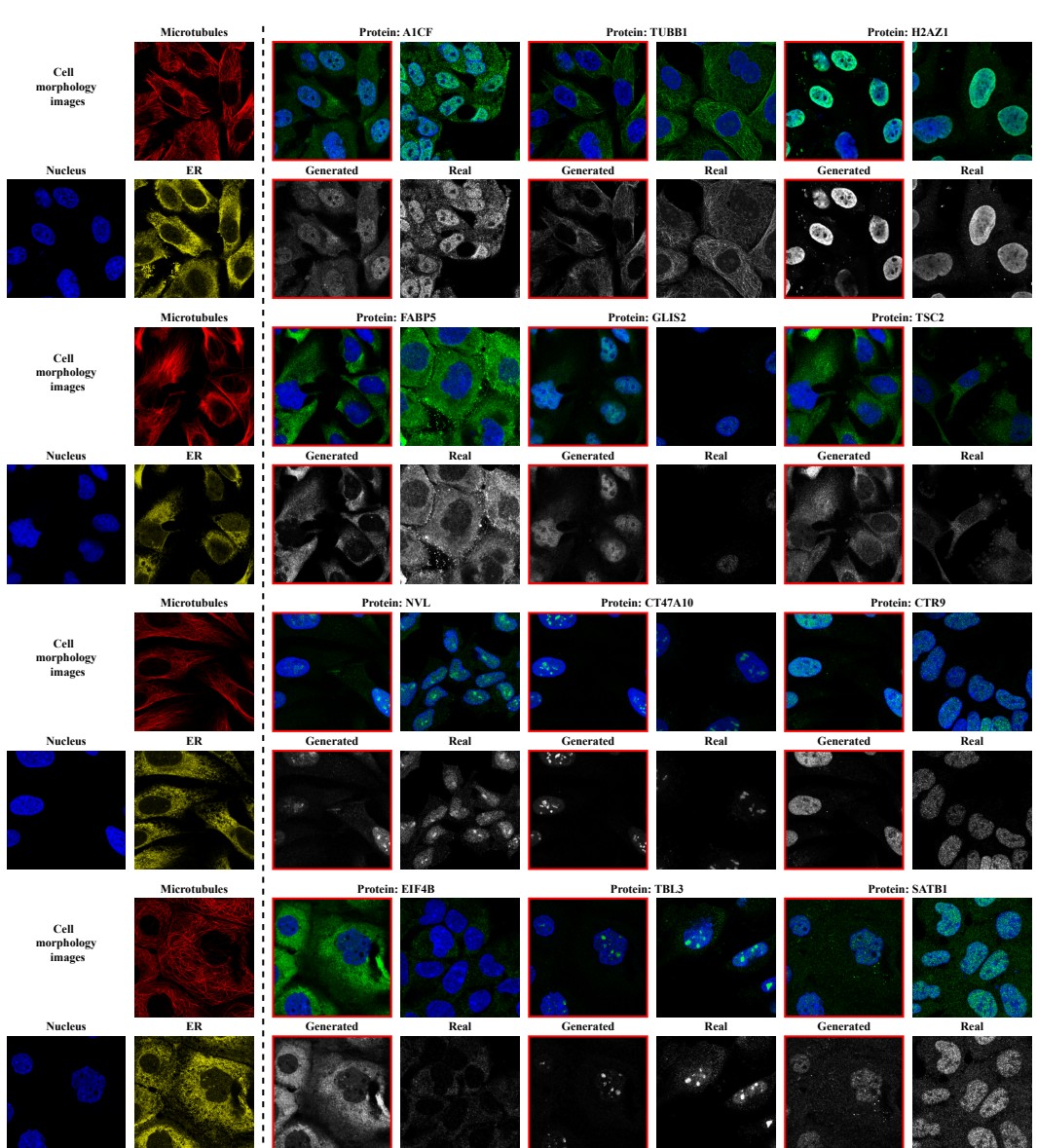

Figure 9: Virtual staining using HPA dataset.

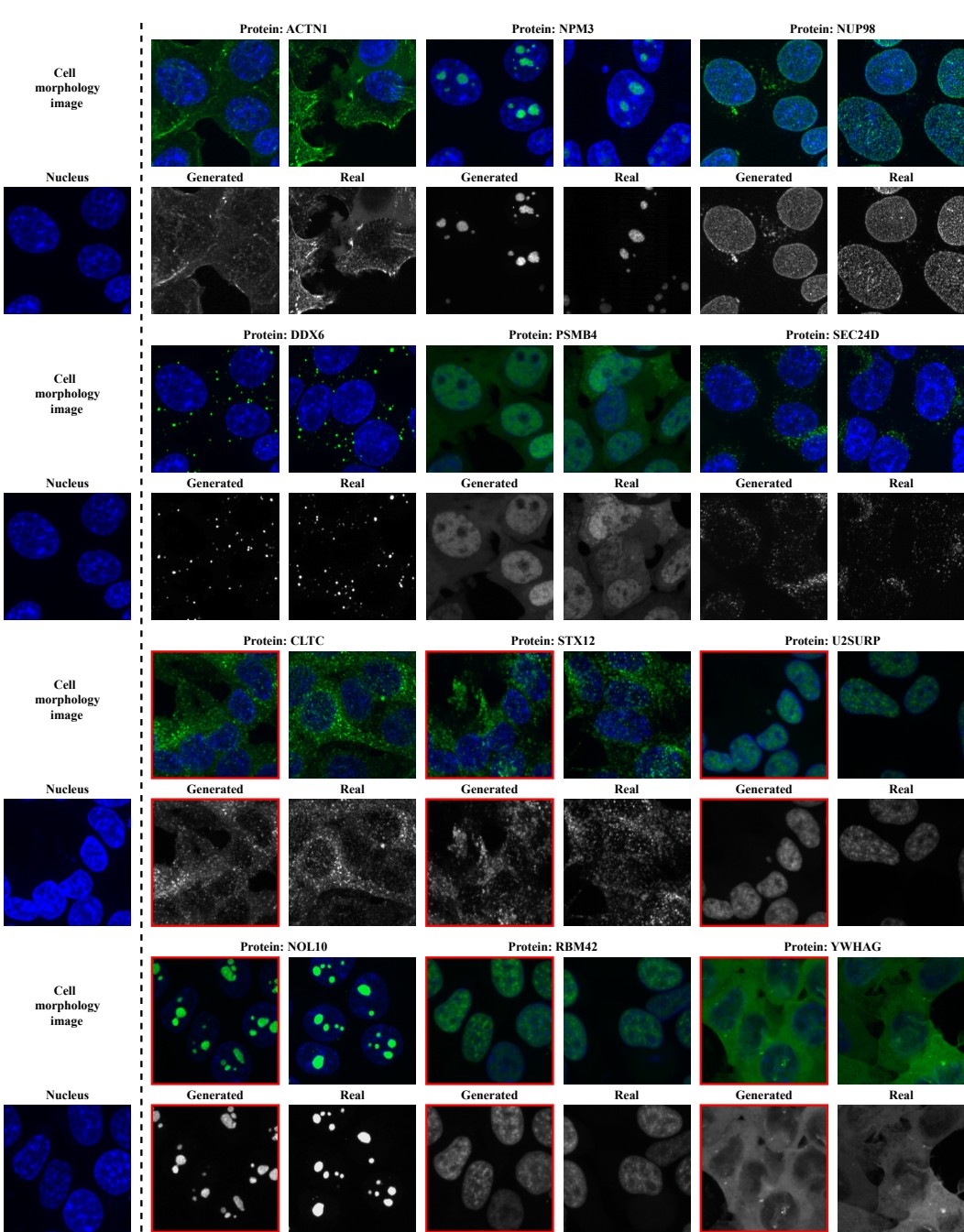

Figure 10: Virtual staining using OpenCell dataset.

Table 3: Generated NLS sequences.

| Index | N-terminus | C-terminus |
|---|---|---|
| 1 | AKSEK | PSPFVM |
| 2 | KKVES | LVTLAERP |
| 3 | KNPTDS | LVKLAERD |
| 4 | ENFTAS | LPALAERR |
| 5 | ENPTAR | GVKLAERD |
| 6 | ANLTAS | PVKLAERK |
| 7 | ENRTAR | FIGVFPGGFIF |
| 8 | ENRTDS | PIAFDRMKFIL |
| 9 | KNRTAL | AVPVEEGDEKFQE |
| 10 | ENGGAS | AKGQLEGDLKFEE |
| 11 | SDNSSAGF | KKKVLKGDLKFEK |
| 12 | SDIERIAEK | LRPVAEGGEKFEE |
| 13 | KKKERIEQF | RTFLRPPKVKMEQRE |
| 14 | KSCERQYLF | RTFLRPPKVKMEERE |
| 15 | GDIDCSEKF | RTFLRPGKVKMEQRE |
| 16 | SDRERITTH | IEKIKRPRSLNAETKY |
| 17 | IIFDPGRQKRLKK | IEKIKRPRSSNAETEY |
| 18 | ASFYETRYERLTN | IEKIKRPRSSNAETVR |
| 19 | LAFIAGRGERKKK | IEKIKRPRSSNAETWY |
| 20 | CIFDIRQKTRLIN | IEKIKRPRSSNAETLY |
| 21 | SSLLKVDQEVKLKVDS | IEKIKRPRSGNAETLY |
| 22 | SDLLKVDQEVKKKVDS | IEKIKRPRSSNAETDY |
| 23 | SSLLKVDQEVKLKVDR | IEKIKRPRSLNAETLY |
| 24 | SSLLKVDQEVELKVDS | YAKEELEEEDESDDDNM |
| 25 | SSFEICRLVFLVFGMLCPA | NVKYCRENPLEEPESPIAKTK |
| 26 | SKNDVIRLQRKRPGVSRDPEM | NVRARIVNGLEVEENPSNKLE |
| 27 | PLIEVLREAVGRSGVRRDYYE | KSRACVTNTPEGEASILNSLL |
| 28 | SKVNRVTTVRERKGVRYVSNE | KVYACRPWKFEERESNLNKAE |
| 29 | IDVSVLDLNFGKTGVRYDYHI | DKYACRDLNFRKEECRYNKTI |
| 30 | PLNNVQRLHVEERGHRLDYAN | GKSACRYNKGDNLDIDNLVLE |
| 31 | GDLDVSYTFRERMDVRYDYEE | GAYYCKSSKGGGKKCAGKKEKK |
| 32 | PLNERNEGQRGRPGVRIVYYY | GKKYQVSSKGGGDKSALKVEKK |
| 33 | PDVNRIELGVLRDDVHLVYHE | GKKLQVSNKHDGKKCALKKELK |
| 34 | PQNEYIEEHRKRYDLYLVYGEK | GKYYRVSSNPEGKKCINKPLLK |
| 35 | PQNEYQEEVRGRTDLRLVYGER | GVYYCVSNKPGGKKCAALKEKK |
| 36 | PQNELIEEVRKRYWYRLDLGVR | GAKYQVTSSPEGKKCANHPLPK |
| 37 | PQNEYQEEVRGRYDLYLVKGEK | GKKYQKSSSGGGDKCILLKEKK |
| 38 | PLNEYIELVFKRADYRRDLHEK | EAYYQVSSKSEGKKCILLKEKV |
| 39 | GLLEYIEEVRGRADLYLVLHER | GAKYQVSSKGENKKSINEKEKK |
| 40 | VLNEYEEEVRGTYDYRRVLHPK | GAYYCVSSNPTGQKCINAVEKK |
| 41 | PQNELIEQVFGRYDYRLDYGEK | QEEAPESELPPELKPKQEEEELQ |
| 42 | PLNEYIELHRKRYDYRFVYWLK | EVKEDEPELKREEIEKATKELDS |
| 43 | GQNDYCELVRGSADFRRVLGVR | YKEAEEYKLKYYLAPKHTEEIDS |
| 44 | PPATDSQKSIISPVINHYKFIYS | RPQQKAQPAQPADEVAEKADEPMEH |
| 45 | PERPDESETNPSLVLRASSDELT | RPQRKAQPAQPADGPAEKADEPMEH |
| 46 | RRNQYDNDVTVWSPQGRIHQIEYAM | RPQRCAVPAQPADEPAEKADEPMEH |
| 47 | RRNQYENDVTVWSPQGRIHQIELAM | RPQRKAQPAQPADEVAEKADEPMEH |
| 48 | RENQYDNDVVVWSPQGRIHQIEYAM | RPQRKAQPAQPADEPAEKADEPMEH |
| 49 | ERNQYDNDVTVESPQGRIHQIRYAM | RPQQKAQPAQPTDEVAEKADEPMEH |
| 50 | FRNQYDNDVTVWSPQGRIHQIEYAM | RPQRKAQPAQPADERAEKADEPMEH |

Table 4: Generated NES sequences.

| Index | N-terminus | C-terminus |
|-------|-----------|-----------|
| 1 | VELDPFGAPA | DKDEDEGFN |
| 2 | EELDPFGAPA | ISDKQSMLVH |
| 3 | AELDPFGAPA | ISLKQAPLVH |
| 4 | SSASDAMAKHE | ISVKQAGLVH |
| 5 | ASASGAMAKHE | IHFKQAPEVH |
| 6 | SIFTPTRQIRLT | ISLCFSPLVH |
| 7 | VSWIISYLVVLIFG | ISVKQAYGVH |
| 8 | VSWIISRLVLLIFS | ISLQQAPEVH |
| 9 | VSWIISRLVVLIFG | ISQKQAPEVH |
| 10 | GEFNEKITLCGTVCL | LKDVLEGDEKFE |
| 11 | GNINEKKTTIGEVCV | LKDVEEGDLKFE |
| 12 | GSINEKKTTCGTVCL | TRPKKKTSGGTDSA |
| 13 | SPFNRKSTTCGTVCL | TRPKKKTSGGGDSA |
| 14 | DSWEDLVDQVLGATKNE | IFSKCLYRGHKLEHY |
| 15 | DDREDFVVLKLVANQAE | IFTCCLYGSSKLEHY |
| 16 | DDGEDGDYQAKDAFSAE | IFTCCLYRSAKLEHC |
| 17 | DDIEDLDYQALVAFQAE | IFTCCLYRSLKLEHI |
| 18 | DDWEDIRVQRKLAGQLE | IFTCALARSGKLEHY |
| 19 | DDREHTVYQASLAPMLE | IFTACLYRSLKVEHK |
| 20 | DSRMDSGYDDLLAVQLE | GEEQNLEALQDRIDENL |
| 21 | PSGRPEEAWEAVVGAAER | GEEQNLIALQDVLDDNQ |
| 22 | PSGRPEELWEAVVGAAER | GEEQNIEAVQDSFDENQ |
| 23 | ASGRPLELWEANVGAAER | AEEQNKEAIQDVEDENQ |
| 24 | SSGRPEELWEAVVGAAER | IPRPRSNTSDGQKLKGKT |
| 25 | ASGRSEELWEAVKGAAER | LPRPRLNASDFQSLKSTY |
| 26 | PSGRPFELWEAKVGAAER | LPRPRLNTSDFQELKPKA |
| 27 | ASGRPHELWEAVVGAAER | LPRPRLNISDFQKLKLVY |
| 28 | ESGDPRELWEAVVGAAER | LPRPRLNTSDFQLLKRKE |
| 29 | ESGRPPELWEAKVGAAEN | LPRPRLNKSDFQSCKPKI |
| 30 | SSEENCRLVVLVFGMCCPA | IPRPRLNASDFQSLKKGY |
| 31 | SSEMILRLVVLVTGMSCPA | LPRPRTCISDFQKFKEKV |
| 32 | HRRGVARGAIAKKKLAELKY | IPRPRLNTSDFQKLKRKG |
| 33 | HLRGVGAGAIAKKKLIEAKY | IPRPRLNTSEFQELYMKE |
| 34 | VTRGVGRGAIADKKLAEAKY | LFTDDYSQEITAEHYREALK |
| 35 | HRRGVRMGAIAKKKLAEAKY | LFTDLYSQKITAEEARELLK |
| 36 | HRRGVNMGAIAKKKLAEAKY | LFTDLVSQAITAEEARDDLP |
| 37 | HRRGVGKGAIAKKKLAEAKY | LFTDLYSQEITAEEPREAAP |
| 38 | HSRCVGGGAIADKALAEDKY | LFTDLYSQEITAEEARELLN |
| 39 | HLREVLGGAIAKKKLAEAKY | LFTDLYSNCITFEEYREDLP |
| 40 | HEREVGLGAIAKKKLAEAKY | LFTDLYSQEIGDEEYREALP |
| 41 | HFREVGAGAIAKKKLAELKY | LFYDLYSQEITKEEPREALK |
| 42 | ALAKLLLESNIRLWVNRPSIIIT | LFTDLYSQEITKEEPREALK |
| 43 | ALPRNLLKSSIVPWVNISSVIQT | LFTDLYSQPITDEEPREALK |
| 44 | ALEEALLFSSLVSWNVYPIVIQK | NKPMAKDKEGFTMYKYILQHKIQ |
| 45 | AVDKNGLNGNIREVNVIPIIIIT | NLLMPTDLAKIGPHWRSLDTSSS |
| 46 | SGPKDMLELGGVIWNNRSQNLYS | EAIMLISIDEGNEFKAELNGKTV |
| 47 | AGPKNLLELGIVLVVRLYKFILS | MSGAPDTLGQGGGGGGGGGPGSGR |
| 48 | ALNEGLLEGLGQLVVQTVSNIYK | MSGAPDTLGQGGGGGGGGGGGSGR |
| 49 | AAASAGATRALLLLLMAVAAPSRA | MSGAPDTGSQGGGGGGIGGYGSGR |
| 50 | AAASAGATRWLLLLLMAVAAPSRA | MSGAPDTLGQGGGGGGGGGGTGSGR |

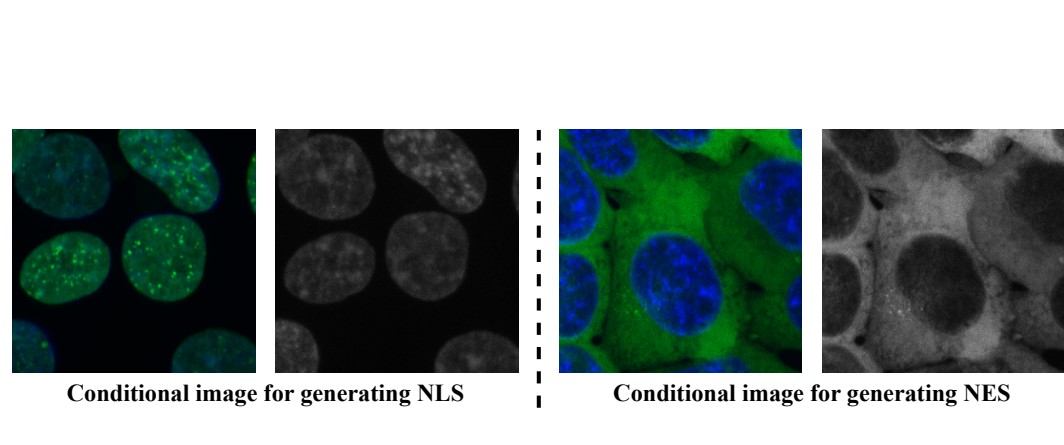

**Conditional image for generating NLS**  **Conditional image for generating NES**

Figure 11: Conditional images for protein localization signal generation.

