# OpenReview forum: "CELL-Diff: Unified Diffusion Modeling for Protein Sequences and Microscopy Images"
_ICLR.cc/2025/Conference — ICLR 2025 Conference Withdrawn Submission_

### Official Review · Reviewer_oAYQ · 2024-10-30

**Soundness:** 3
**Presentation:** 3
**Contribution:** 3
**Rating:** 5
**Confidence:** 4

**Summary:**

The authors present Cell-Diff, a bidirectional diffusion-based method capable of generating microscopy images from protein sequences and reconstructing protein sequences from microscopy images. They combine a continuous diffusion model for generating microscopy images with a discrete diffusion model for reconstructing protein sequences into a single pipeline. CELL-Diff was evaluated on two datasets against a transformer-based model, CELL-E 2, showing improved qualitative and quantitative performance.

**Strengths:**

1. **Potential Impact:**

Applications such as virtual staining and the ability to computationally screen protein localisation signals demonstrate the model's practical relevance for biological discovery.

2. **Technical Depth:**

The paper provides a detailed breakdown of continuous and discrete diffusion processes, particularly in the context of microscopy images (continuous) and protein sequences (discrete). Integrating these diffusion models into a unified framework is well-motivated and grounded in previous work. The implementation of cross-attention and the use of the ESM2 model for sequence embeddings enhances the model's capacity to handle multimodal data.

**Weaknesses:**

1. **Technical clarity:**

While the diffusion framework is well-documented, there are moments where clarity is lacking, particularly when describing how the continuous and discrete diffusion models are integrated. It would be beneficial to have more explicit guidance for readers unfamiliar with diffusion models applied to biological data. The equations provided, though rigorous, could benefit from more explanatory commentary. For example, the problem description is not clear. The masking of the protein sequence and prediction of this is not clear from the beginning, and they are assumed to be known. It should be stated that you are predicting letters in a masked protein sequence in the training procedure (if this is indeed what is happening).

2. **The table and figures are not clear**:

- Table 1 is difficult to read. There should be a separation between the methods.
- Figures 4, 5, and 6 are difficult to follow. Each row and column should be labelled or described in the caption.

3. **Maximum Spatial Frequency (MSF) is very unclear:**

- The authors briefly introduce their Maximum Spatial Frequency (MSF) resolvability metric in the paper, but they do not fully explain this. Furthermore, the definition is not clear.
- What is the unit of measurement $nm$? Are these nanometers? If so, why is it measured in nanometers? I assume that this comes from the pixel size; however, this is not clear.

4. **Lack of sequence generation:**
- While the authors mention bidirectional generation between protein sequences and microscopy images in the abstract and introduction, the paper primarily focuses on generating protein images from sequences and does not show any practical implementation or experiments that involve generating protein sequences from images. While the results of sequence generation are in the appendix, they should be moved to the main text.

5. **Minor:**
- Grammatical and spelling errors: There are a few grammatical errors which should be corrected
	- “which involves model a continues variable” in line 210.
	- The protein image is first defined as $I^{prot}$ in line 207. Then, from line 215, this changes to $I^{port}$ in the rest of the paper.
	- “with 100 steps to accurate the sampling speed” in line 400.
- Incorrect in-text citation:
	- For example, in line 305 “using a pre-trained ESM2 model Lin et al. (2022).” 'Lin et al. (2022)' is formatted as a narrative citation, which would typically be written with \citet{}. However, since it follows the phrase 'using a pre-trained ESM2 model,' it would be more appropriate to use \citep{} for a parenthetical citation (i.e., '(Lin et al., 2022)'). Adjusting these citation styles to align with the context will improve the clarity and consistency of references throughout the paper.
	- Other examples are in line 376 “ Adam optimizer Kingma & Ba (2014)” and in line 420: “we compute the Fr´echet Inception Distance (FID) Heusel et al. (2017)”.
- Avoid using the word “significantly” if no statistical significance tests are done. Use “considerably” instead.

**Questions:**

1. Given the fact that the compared algorithm CELL-E 2 produces binary image outputs, how fair is using MSF to evaluate the models? Could the authors please discuss the potential limitations or biases of using MSF for comparison when one model produces binary outputs and the other doesn't?

2. The balancing coefficient $\lambda$ is introduced in equation 12 and is set to 100 in the experiments. How does the value of $\lambda$ affect the results? Could the authors please provide an ablation study showing how different values of $\lambda$ impact the model's performance across various metrics?

3. What do the authors mean by this in line 418: “To calculate IoU, we apply median value thresholding to the original protein images to generate binary masks, while for CELL-E2, we use the predicted thresholding image"? Do you mean that median value thresholding was applied to both the original and generated protein images to obtain ground truth and predicted masks for IoU? Furthermore, CELL-E 2 generates masked (binary) images.  When assessing the FID-O, is it an unfair comparison for CELL-E 2? If so, this should be stated clearly. Please provide a step-by-step explanation of how IoU is calculated for both models, and discuss any potential biases in the FID-O comparison due to the differences in output types between the models.

4. From Table 1, CELL-E 2 outperforms CELL-Diff in terms of IoU when using the Nucleus image alone as input. However, it is not tested how CELL-E 2 performs when using the additional channels. Could this method be tested with these additional channels?

---

### Official Review · Reviewer_ftJJ · 2024-10-30

**Soundness:** 2
**Presentation:** 3
**Contribution:** 2
**Rating:** 5
**Confidence:** 3

**Summary:**

The paper introduces CELL-Diff, a unified diffusion model that enables bidirectional generation between protein sequences and their corresponding fluorescence microscopy images. By integrating continuous diffusion models for images and discrete diffusion models for sequences within a unified framework, CELL-Diff can generate detailed protein images conditioned on protein sequences and cell morphology images, and conversely, generate protein sequences based on microscopy images. The model employs an attention-based U-Net architecture and is trained on the Human Protein Atlas (HPA) dataset, with fine-tuning on the OpenCell dataset. Experimental results demonstrate that CELL-Diff outperforms previous methods like CELL-E2 in generating high-fidelity protein images with finer details, making it a valuable tool for investigating subcellular protein localization and interactions.

**Strengths:**

1. The paper presents a novel approach by integrating continuous diffusion models for images and discrete diffusion models for sequences into a unified framework. This integration allows for efficient bidirectional generation between protein sequences and microscopy images.
2. CELL-Diff achieves higher resolvability in generated images compared to previous methods like CELL-E2. The authors report that CELL-Diff's generated images approach the resolvability of original training data, enabling discernment of finer details in protein distribution.
3. The paper demonstrates potential applications of CELL-Diff in virtual screening of protein localization signals, virtual staining to overcome fluorescence microscopy limitations, and generation of novel protein localization signals.
4. The authors conduct thorough experiments on both the HPA and OpenCell datasets, including quantitative metrics like MSF-resolvability, IoU, and FID scores, as well as visual comparisons, to validate the effectiveness of CELL-Diff.

**Weaknesses:**

1. While the model is trained and fine-tuned on HPA and OpenCell datasets, the paper acknowledges the size limitations and lack of diversity in protein sequences. This could affect the generalizability of CELL-Diff to unseen protein sequences or different cell types.
2. The paper primarily compares CELL-Diff with CELL-E2. Including comparisons with a broader range of existing models or baselines could strengthen the validation of CELL-Diff's performance.
3. While potential applications are presented, the paper could provide more in-depth analysis on how CELL-Diff's generated outputs correlate with biological reality, including potential pitfalls or limitations in practical biological interpretations.

**Questions:**

1. The paper describes that CELL-Diff can generate protein sequences based on microscopy images. However, it is not clear how the model ensures that these generated sequences are biologically valid and functional. Given that small changes in amino acid sequences can have significant effects on protein function, how does the model account for this? Are there mechanisms in place to validate the functionality or structural stability of the generated sequences?
2. Proteins often undergo post-translational modifications (PTMs) that affect their localization and function. Does CELL-Diff consider PTMs in the generation process? Additionally, how does the model perform with proteins that have complex quaternary structures or those that form part of larger protein complexes?
3. The paper mentions that "the variability in equipment and experimental conditions limits the availability of robust image encoders". How does CELL-Diff address this variability in the training data? Does the model generalize well to images obtained under different imaging settings, and is there any preprocessing required to standardize the inputs?
4. The conditional generation relies on cell morphology images such as nucleus, ER, and microtubule markers. How sensitive is CELL-Diff to the choice or quality of these images? For instance, if only partial or noisy cell morphology images are available, how does that affect the quality and accuracy of the generated protein images?
5. Has the potential incorporation of other types of biological data (e.g., gene expression profiles, protein-protein interaction networks, or 3D structural data) been considered to enhance the model's capability? How might the addition of such multimodal data affect the training complexity and the interpretability of the results?
I would be pleased if the author could consider raising my score after addressing my concerns.

---

### Official Review · Reviewer_5Pdu · 2024-10-31

**Soundness:** 1
**Presentation:** 3
**Contribution:** 2
**Rating:** 5
**Confidence:** 5

**Summary:**

The manuscript entitled "CELL-DIFF: UNIFIEDDIFFUSIONMODELING FOR PROTEINSEQUENCESANDMICROSCOPYIMAGES" proposes a bidirectional generative model for generating protein sequences from fluorescence micrographs and vice versa. The authors present a novel and interesting approach to what appears to be bidirectional virtual staining, by tying it to the sequence of proteins. However, oddly the authors appear to have made a set of unrealistic assumptions and simplifications while making unsubstantiated claims. While I genuinely like the interdisciplinarity of the contribution, the specific weakness points must be addressed and claims have to be revised.

**Strengths:**

Authors propose a novel bidirectional discrete-continues diffusion model for image and sequence data.

The authors present an interesting idea to connect genotype and phenotype.

**Weaknesses:**

n of protein exists in databases like DAVID (NIH), however, I haven't seen any mention of this source here.

Secondly, microscopy data is very poorly described in this work, essentially taking it at face value. For example, the field of view of each micrograph may contain multiple cells, a single Z slice, a maxima projection of confocal slices or a total projection of epifluorescence. Furthermore, all of this is convolved with a PSF of an imaging system. Quantum yield of the fluorophore, camera exposure time, PMT gain, magnification, NA, laser power, imaging medium, and immersion are just a number of conditions that need to be standardised to make the image quantitatively comparable. In other words, unless conditions are identical, two micrographs are incomparable. If these conditions were standardised in some way - this should be discussed. This results, in reality, in the fact that many micrographs of the same protein appear different while on other occasions micrographs of different fluorescent proteins would appear the same. Take for example fluorescence micrographs of transgenic fluorescent Nucleoplasmin and Histone 2B. At the magnification level shown in Figs 1-3 both would look like a nuclear stain. In this case, even if there’s an inductive bias present allowing connecting them to their sequences, there’s very little information about their function in micrographs and sequences combined. In principle, you can try to fit anything to anything, but in the absence of a solid inductive bias, you are just going to force the model to hallucinate.

Thirdly, when it comes to fluorescence, the actual source of the signal is not the entire protein. In the case of chimeric proteins, it is a fluorescent insert. In the case of immunohistochemistry, it's the fluorescent dye of the secondary conjugate. Assuming the former was the setting here if the sequence contains the chimeric insert you are just fitting images to a handful of fluorescent protein subsequences: CFP, GFP, RFP/mCherry - there are not that many. Please discuss this.

**Questions:**

What images did you use? Please describe the exact conditions, modalities, equipment, and setting. Make sure to note if the images are quantitatively comparable. If not what steps did you take to make them quantitatively comparable?
What was the inductive bias? Showcase exact proteins and morphology or actual function as controls. What are the conditions when it works, and what are the conditions when it fails?
What was the molecular source of fluorescence?
Were these chimeric proteins?
Did sequences contain the fluorescent protein parts? If so this constitutes a data leakage. How did you mitigate this?

---

### Official Review · Reviewer_qVw3 · 2024-11-02

**Soundness:** 3
**Presentation:** 2
**Contribution:** 2
**Rating:** 3
**Confidence:** 4

**Summary:**

The authors utilized over 88,000 samples of (protein sequence, protein image, nucleus image, ER image, microtubule image) and over 6,000 samples of (protein sequence, protein image, nucleus image) to train and fine-tune a hybrid continuous (for images) and discrete (for protein sequences) bidirectional diffusion model. This model translate images and protein sequences.  They tested their method, CELL-Diff, on 100 proteins after training on the remaining ~1000.  CELL-Diff was compared to CELL-E2 (2024) using several metrics: a Fourier Ring-based evaluation of discerning fine image structures; an Intersection over Union (IoU) on thresholding-generated masks; as well as two version of the Fréchet Inception Distance (FID), one applied to the thresholded images (FID-T) and one on the original images (FID-O). The method’s utility is demonstrated through three biological applications: virtual screening, visual staining, and localization of protein sequences.

**Strengths:**

- The work is ambitious given the complexity of the task.

- The methodology of bidirectional diffusion model for translating between proteins sequences and protein images appears to be sound.

- Various evaluation metrics and potential biology application are presented.

**Weaknesses:**

- The utility and trustworthiness of such predicted data (especially in critical biomedical applications, e.g. drug design) is questionable, particularly when there are clearly noticeable differences between ground truth and simulated data.

- Comparisons has some issues as discussed below.

- I believe the results of this work should be assessed by biology experts. I don't think ICLR is the right venue for this work.

- The work synthesizes protein images and sequences. Although these images may 'look like' real images and sequences, they may contain wrong information that is not grounded in a real biological phenomenon. Thus such synthetic data should not be trusted, as this could cause harm, for critical biomedical applications.  The authors should make a compelling case why there is a need to produce such synthetic data and should clearly mark synthetic images and data as such in every figure.

**Questions:**

L180: Delete the word "where".

L210: Typo: "involves model a continues" --> involves modelling a continuous.

L215: Typo: L^{port} --> L^{prot}.

Fig. 2 and upper panel of Fig. 3 and their captions are clear. It would help the reader if the different parts of figures 2 and 3 are named (a, b, c, ...), and then referenced in the text where appropriate as Fig2a, Fig2b, Fig3a, etc.

Fig3: Clarify and identify the lower panel of Fig. 3 (below the dashed line) and describe it in the caption, and describe how it is related to the top panel, and related to the overall method?

Eq. (13):  small i is not (explicitly) defined.

Eq. (13): it unclear how to apply this equation. How is the f=I/(image size x pixel size) related to the "where {FRPSD...}" part?

L418: You write that the thresholding is applied of the original protein images. Clarify that/whether the thresholding was also applied to the generated (predicted) protein image.

The authors do not explain why they extract the masks (via thresholding) and apply IoU, noe they do explain why they measure FID limn the masks (i.e. FID-T).

What does the column "Cell image" mean in Table 1? How were these images used; and why aren't the same entries listed next to the competing method CELL-E2 (e.g. "Nucl,ER" vs only "Nucl")?

The bolded 'winning' values in table 1 should be restricted to the top two rows. The rows 3,4,5 (Cell-Diff with ER or/and MR) should be isolated from the other results.

Table 1:

* The way this table is presented raises questions about the fairness of comparison, i.e. could the improved results in Table 1 - HPA (the bold values) be attributed to using more/richer types of training data (ER and MT) and not to the methodology itself?  The authors do state, on lines L430-431, that the competing methods CELL-E2 and  the proposed method CELL-DIFF achieve comparable results, and that [only] when other images are used (ER and microtubules) do the proposed method performance surpasses the competition.

* The bolded 'winning' values in Table 1 - HPA should be restricted to the first two rows (CELL-E2 Nucl vs CELL-Diff Nucl), and the comparison (bold text) should be based on these two rows.

* The rows 3,4,5 (Cell-Diff with ER or/and MR) should be isolated from the other results and the values for those rows should not be used in the comparison. Examining the first 2 rows of results in Table 1, we see that CELL-E2 has better IoU than CELL-Diff (0.461 vs 0.448).

* How do explain that using ER and MT images give worse FID-T then when using only ER (34.1 vs 32.4; in Table 1, HPA, FID-T)?

Figure 4 show that CELL-Diff produced sharper images and qualitatively images that are more similar to the real images than CELL-E2. However, it is noticeable that there are structures / pixels that are clearly not faithfully reconstructed. Perhaps this localized differences are critical for a particular application. Similar observations can be made on the results in the Appendix (Fig. 7 and Fig. 8). So this raises the question of how would one trust the predicted images and utilize them for subsequent tasks? The authors do mention different potential downstream applications.

Whether the CELL-Diff images do in fact perform better when used in some subsequent tasks when compared to CELL-E2 remains to be seen. For example, how will the results look like if a figure like Figure 5 is produced but for CELL-E2?

The competing method CELL-E2 is not compared against under the "virtual screening" application.

The "visual staining" application is assessed qualitatively (again without comparison to CELL-E2). For this application, the authors write: "we solve this problem..."; I don't believe the authors provided anywhere close to sufficient evidence to justify the use of these words.  The authors also state "clearly shows the association of LIF18A and EML4...  consistent with known biological function" - a reference is needed here.   Further, the overlaid image in Figure 6 does not, at least to the reviewer, clearly show the association. Other better ways to clearly visualize, and even quantify, the association is warranted.

For the "localization signal generation", the generated sequences are reported in a table in the appendix, but there is no assessment of their correctness.

**Details Of Ethics Concerns:**

Please see the last point under Weaknesses.

---

### Note · Authors · 2024-11-22

I have read and agree with the venue's withdrawal policy on behalf of myself and my co-authors.